# Multisensory Stimulation Reverses Memory Impairment in Adrβ_3_KO Male Mice

**DOI:** 10.3390/ijms241310522

**Published:** 2023-06-23

**Authors:** Thaís T. Ravache, Alice Batistuzzo, Gabriela G. Nunes, Thiago G. B. Gomez, Fernanda B. Lorena, Bruna P. P. Do Nascimento, Maria Martha Bernardi, Eduarda R. R. Lima, Daniel O. Martins, Ana Carolina P. Campos, Rosana L. Pagano, Miriam O. Ribeiro

**Affiliations:** 1Programa de Pós-Graduação em Distúrbios do Desenvolvimento, Centro de Ciências Biológicas e da Saúde Universidade Presbiteriana Mackenzie, São Paulo 01302-907, SP, Brazil; thaisterpins@hotmail.com (T.T.R.); alice.batistuzzo@gmail.com (A.B.); gabriela.nunes96@outlook.com (G.G.N.); gneccao@hotmail.com (T.G.B.G.); fee.lorena@gmail.com (F.B.L.); bruna.pascarelli@gmail.com (B.P.P.D.N.); 2Departamento de Medicina Translacional, Universidade Federal de São Paulo 04023-062, SP, Brazil; 3Graduate Program in Environmental and Experimental Pathology, Paulista University, São Paulo 04026-002, SP, Brazil; marthabernardi@gmail.com; 4Laboratory of Neuroscience, Hospital Sírio-Libanês, São Paulo 01308-050, SP, Brazil; eduardarosolem@gmail.com (E.R.R.L.); daniel.martins@hsl.org.br (D.O.M.); anacarol.pcampos@gmail.com (A.C.P.C.); rosana.lpagano@hsl.org.br (R.L.P.); 5Sunnybrook Research Institute, Toronto, ON M4N 3M5, Canada

**Keywords:** noradrenaline, cognitive benefits, memory, aging, environmental enrichment

## Abstract

Norepinephrine plays an important role in modulating memory through its beta-adrenergic receptors (Adrβ: β_1_, β_2_ and β_3_). Here, we hypothesized that multisensory stimulation would reverse memory impairment caused by the inactivation of Adrβ_3_ (Adrβ_3_KO) with consequent inhibition of sustained glial-mediated inflammation. To test this, 21- and 86-day-old Adrβ_3_KO mice were exposed to an 8-week multisensory stimulation (MS) protocol that comprised gustatory and olfactory stimuli of positive and negative valence; intellectual challenges to reach food; the use of hidden objects; and the presentation of food in ways that prompted foraging, which was followed by analysis of GFAP, Iba-1 and EAAT2 protein expression in the hippocampus (HC) and amygdala (AMY). The MS protocol reduced GFAP and Iba-1 expression in the HC of young mice but not in older mice. While this protocol restored memory impairment when applied to Adrβ_3_KO animals immediately after weaning, it had no effect when applied to adult animals. In fact, we observed that aging worsened the memory of Adrβ_3_KO mice. In the AMY of Adrβ3KO older mice, we observed an increase in GFAP and EAAT2 expression when compared to wild-type (WT) mice that MS was unable to reduce. These results suggest that a richer and more diverse environment helps to correct memory impairment when applied immediately after weaning in Adrβ_3_KO animals and indicates that the control of neuroinflammation mediates this response.

## 1. Introduction

It is well established that norepinephrine (NE) plays an important role in modulating memory consolidation [1,2] in mammals through the activation of beta-adrenergic receptors (Adrβs) expressed in the hippocampus (HC) and amygdala (AMY) [3,4]. The noradrenergic system is known to strengthen long-term potentiation (LTP) within the dentate gyrus of rats during arousing experiences, such as exposure to novelty [5,6]. This also drives neuronal activity in the locus coeruleus (the main point from which noradrenergic neurons project throughout the brain) and is partially blocked by inhibition of Adrβs [7,8]. Although the role of Adrβ1 and Adrβ2 is well established, recent experimental evidence has indicated that Adrβ 3 has a key role in mediating memory consolidation in rodents [9,10]. Confirming this, the administration of an Adrβ_3_ agonist has been shown to reverse memory impairment in animal models of Alzheimer’s disease [11].

There is extensive evidence that multisensory stimulation (MS) profoundly affects animal behavior. Specifically, heightened sensory stimulation and problem-solving opportunities enhance performance in various learning and memory tasks [12,13,14]. MS encompasses different types of stimuli, such as physical, nutritional, sensorial, cognitive and social [15]. MS is known to increase neurogenesis [16,17] and has been shown to improve learning and memory consolidation in several animal models [18,19,20,21,22]. Eight weeks of MS was shown to significantly improve impaired mood and cognition and reduce levels of anxiety and depression in adult male offspring of hypothyroid rat dams [12]. Three years of music training reversed the reduction in the size of the HC in children with congenital hypothyroidism [13]. 

The activation of astrocytes and microglia can cause cognitive decline and memory impairment (in part or totally), effects that can be identified by the upregulation of glial fibrillary acid protein (GFAP) and ionized calcium-binding adapter molecule 1 (Iba-1) [14,15,16]. Reactive astrocytes have been increasingly recognized as a key contributor to the progression of many neurodegenerative diseases [17]. Deletion of astrocytic Excitatory amino acid transporter 2 (EAAT2), the major glutamate transporter in the brain, leads to early deficits in short-term memory and in spatial reference learning and long-term memory [18]. Considering that the inactivation of Adrβ_3_ induces a significant impairment in short- and long-term memory [10] and that neuroinflammation controls it [19], we hypothesized that the use of MS could reverse the memory impairment exhibited by Adrβ_3_ knock-out (Adrβ_3_KO) mice. We found that the cognitive impairment exhibited by Adrβ_3_KO mice at 120 days of life was rescued following an 8-week program of MS initiated after weaning at 21 days of life; however, when the same 8-week MS program was started at 120 days of age, the cognitive impairment persisted. 

## 2. Results

### 2.1. Ambulatory and Exploratory Activity of Adrβ_3_KO and WT Mice Exposed to MS Early in Life

MS exposure did not affect ambulatory activity in an open-field (OF) test (Figure 1A) but decreased the time spent in the periphery of the OF in both WT and AdrβKO mice on day 3 of observation (Figure 1B). The MS protocol reduced exploratory behavior only in the AdrβKO mice on day 3 of observation (Figure 1E). Bonferroni’s comparisons test showed reduced exploratory activity in the ARβ3KO-MS group relative to the ARβ3KO group (*p* = 0.02).

### 2.2. MS Exposure Early in Life Corrects Cognitive Impairment in Young Adult Adrβ_3_KO Mice

Cognition was evaluated through the novel object recognition test (NOR) and the valence-based social recognition test (SR). The NOR test uses the preference for novelty exhibited by the rodents, and if they spend more time exploring the novel object, it means that they remember the object to which they were previously exposed. In our study, the test evaluated short-term memory (3 h) and long-term memory (24 h). SR refers to the ability of animals and humans to discriminate between a familiar and unfamiliar conspecific and is also used to assess memory in rats and mice. Additionally, it does not require the application of additional stimuli to provoke the response. It is used as an index for memory performance [23]. 

In the NOR test, all groups explored the two identical objects (O1) similarly during the familiarization period (Figure 2A). Three hours after the familiarization period (Figure 2B), the mice were exposed to the O1 and to a new object (O2). Bonferroni’s multiple-comparisons tests showed that the absence of Adrβ_3_ impaired short-term memory, but the MS protocol rescued the performance of the Adrβ_3_KO animals, with increased time spent with the new object (O2). The WT and the WT MS mice exhibited preserved short-term memory (Figure 2B). Twenty-four hours after the familiarization period, Bonferroni’s multiple-comparisons test showed that the absence of Adrβ_3_ affected long-term memory as the mice spent a similar amount of time with the known (O1) and the unknown object (O3) (Figure 2C). The MS protocol improved the ability of the Adrβ_3_KO mice to remember O1 as they spent significantly more time with O3 (Figure 2C). The WT mice retained long-term memory with or without the MS protocol. 

In the SR test, all groups explored the empty cups similarly during the familiarization period (Figure 2D). When exposed to an empty cup and to a conspecific mouse, all groups preferred to spend time exploring the cup with the conspecific mice, showing that the absence of Adrβ_3_ does not impair socialization behavior (Figure 2E). In the social discrimination test (Figure 2F) WT, WT MS, Adrβ_3_KO and Adrβ_3_KO MS mice spent significantly more time with the unknown mice than with the known mice (Figure 2F). The difference in the performance of the Adrβ_3_KO mice in the NOR test compared to the SR test is explained by the fact that the SR test uses conspecific animals, thus memory formation is strengthened by stimulus valence.

### 2.3. MS Protocol in Early Life Decreases Glial Cell Activation

To evaluate whether the cognition impairment associated with the absence of Adrβ_3_ was due to glia activation, we measured the expression of GFAP and Iba-1 in the HC by Western blot. As can be seen in Figure 3A,B, the younger Adrβ_3_KO mice exhibited an increase in Iba-1, but not in GFAP expression. The MS protocol exposure early in life decreased GFAP expression in the HC of both the WT and Adrβ_3_KO mice, while it decreased Iba-1 expression only in the HC of Adrβ_3_KO mice. No alterations were observed in EAAT2 expression (Figure 3C).

### 2.4. Ambulatory and Exploratory Activity of Adrβ_3_KO and WT Mice Exposed to MS Late in Life

Two-way ANOVA analysis showed that the ambulatory activity of the WIT animals when exposed to the OF test (Figure 4A) was not affected by the MS protocol but increased in Adrβ_3_KO MS in the first day of observation (*p* = 0.037). The time spent in the periphery of the OF (Figure 4B) decreased on day 3 of observation only in the Adrβ_3_KO MS adult mice group when compared to the Adrβ_3_KO adult mice (*p* = 0.015). The MS protocol increased the exploratory activity of the WT adult mice on day 1 (*p* = 0.003), day 2 (*p* = 0.02) and day 3 (*p* = 0.0003) of testing (Figure 4C,D). However, MS decreased the exploratory behavior of the Adrβ_3_KO mice on day 1 (*p* = 0.01), day 2 (*p* = 0.02) and day 3 (*p* = 0.02) of testing (Figure 4D–F). Notably, the control Adrβ_3_KO adult mice explored significantly more than the control WT adult mice who were also not exposed to MS on day 2 (*p* = 0.008) and day 3 (*p* = 0.002). 

### 2.5. MS Exposure Late in Life Does Not Correct the Worst Cognitive Impairment Observed in Adult Adrβ_3_KO Mice

In the NOR test, all groups explored the two identical objects (O1) similarly during the familiarization period (Figure 5A). Three hours after the familiarization period (Figure 3B), the mice were exposed to the O1 and to a new object (O2). Bonferroni’s multiple-comparisons tests showed that the older Adrβ_3_KO mice exhibited impaired short-term memory, and the MS protocol was unable to restore or improve this parameter, with a similar amount of time spent with the new object (O2). The WT and the WT MS mice exhibited preserved short-term memory (Figure 5B). Twenty-four hours after the familiarization period, Bonferroni’s multiple-comparisons tests showed that the absence of Adrβ_3_ in the older mice affected long-term memory because the mice spent a similar amount of time with the known O1 and the unknown object O3) (Figure 5C). In addition, the MS protocol did not improve the ability of the Adrβ_3_KO mice to remember O1 because they spent a similar amount of time with O3 (Figure 6C). The WT mice retained long-term memory with or without the MS protocol. 

In the SR test, all groups explored the empty cups similarly during the familiarization period (Figure 5D). When exposed to an empty cup and to a conspecific mouse, all groups preferred to spend time exploring the cup, with conspecific mice showing that aging does not impair socialization behavior regardless of the presence of Adrβ_3_ (Figure 5E). In the social discrimination test, older WT and WT MS mice spent significantly more time with the unknown mice than with the known mice (Figure 5F), but the Adrβ_3_KO mice did not. Exposure to the MS protocol late in life did not restore this behavior regardless of the strength of the stimulus (Figure 5F). 

### 2.6. MS Protocol Late in Life Does Not Decreases Glial Cell Activation

To evaluate whether the impaired cognition observed in the older Adrβ_3_KO mice was accompanied by increased glial activation, we measured the expression of GFAP and Iba-1 in the HC by Western blot. The older Adrβ_3_KO mice did not exhibit alterations in GFAP or Iba-1 expression in the HC (Figure 6A,B). Notably, the MS protocol was not able to reduce GFAP expression, and Iba-1 expression was increased when compared to the Adrβ_3_KO mice (Figure 6A,B). Furthermore, EAAT2 expression was increased in Adrβ_3_KO mice when compared to the WT mice, and the MS protocol was not able to change EAAT2 expression in both the WT and Adrβ_3_KO mice (Figure 6C). Considering that the Adrβ_3_KO mice showed an inability to discriminate between a familiar and unknown conspecific (Figure 6F), we performed analysis of GFAP, Iba-1 and EAAT2 expression in the AMY, a potentially critical site for emotional-processing stimuli, such as that experienced during an encounter with a conspecific mouse [24,25]. As we can see in Figure 7A, there was an increase in GFAP in the AMY of older Adrβ_3_KO mice, and the MS protocol was not able to reduce it. Expression of Iba-1 in the AMY was not affected by genotype or the MS protocol (Figure 7B). EAAT2 expression in the AMY was not affected by genotype but increased in the WT MS and older Adrβ_3_KO MS mice when compared to the WT mice (*p* = 0.02 and 0.034, respectively, Figure 6C). 

## 3. Discussion

The present study revealed that cognitive impairment in younger Adrβ_3_KO mice is reversed by the MS protocol when it is initiated early in life (21 days of age). We also observed that cognitive impairment worsens with aging in Adrβ_3_KO mice, and this deficit was not improved by the MS protocol initiated late in life (120 days of age). Interestingly, glial cells and glutamate transporter expression were not shown to be associated with the cognitive impairment seen in younger Adrβ_3_KO mice, but the MS protocol could decrease GFAP and Iba1 expression in the HC, which might have contributed to reverting Adrβ_3_KO-related cognitive decline. In the HC of the older Adrβ_3_KO mice, there was also no alteration in glial cell expression. After the MS protocol, the older mice exhibited a higher expression of EAAT2, and although studies have demonstrated that increased EAAT2 expression may exert beneficial effects on cognitive function, this was not enough to promote cognitive recovery in the older mice. 

The present results confirm our previous observation that Adrβ_3_KO mice exhibit moderate cognitive impairment [10]. Remarkably, this phenotype was entirely reversed by an 8-week MS protocol when it was initiated in very young mice. We know that MS increases hippocampal neuroplasticity [26,27] and neurogenesis [28,29], but we did not address this process in the present study. It is also known that MS decreases neuroinflammation [30,31], and our data support this important role of MS. We showed that there was a decrease in the expression of GFAP, a marker for astrocyte activation, in the HC of Adrβ_3_KO and WT mice after the 8-week MS protocol. Iba-1, a marker for microglia activation, was also reduced in Adrβ_3_KO mice exposed to the MS protocol. Mounting evidence indicates a close relationship between glial cells and both cognitive impairment [32,33] and the pathogenesis of neurodegenerative disorders, such as Alzheimer’s disease [34,35]. MS has also been shown to produce beneficial effects against inflammation by downregulating the expression of GFAP and Iba-1 in the HC of Adrβ_3_KO and WT mice, leading to reduced glial cell activation and cytokine-mediated inflammation, and this may contribute to improved cognitive functioning and memory [36,37]. NE modulates the activity of microglia and astrocytes [38,39], decreasing the inflammatory markers by binding to β-adrenergic receptors [39,40]. The role of beta-adrenergic receptors in glial cells has been under investigation since the late 1990s, and it has been shown that the Adrβ_2_ present in glial cells can modulate the astrocyte phenotype and phagocytic activity [41,42], while also modulating the activation of classical activated microglia [43]. However, to the best of our knowledge, our study is the first to report glial cell modulation by Adrβ_3_ manipulation, which may shed light on the importance of these pivotal receptors not only for neuronal activation, but also for glial neuroplasticity. 

Older mice exhibited the worst cognition. Younger Adrβ_3_KO mice could discriminate familiar co-specifics and spent more time with the novelty object. Noradrenergic modulation of the AMY is very important for forming emotional memory and in respect of the interaction with an unknown mouse, which involves emotional valence. The older Adrβ_3_KO mice spent a similar amount of time with both familiar and unknown co-specific mice, showing that they did not remember the familiar mouse to which they had been previously exposed. Moreover, the MS protocol initiated later in life did not improve the cognitive deficit exhibited by older Adrβ_3_KO mice. It is possible that if the MS had been initiated by 21 days of age, it could have prevented the deterioration in cognition observed at 180 days of age.

We did not find any alterations in astrocytes or microglia expression in the HC of the older Adrβ_3_KO mice when compared to the WT mice of the same age. A possible explanation for this is the glial modification that occurs in the HC of older rodents. As significant astrocytic modulation has been observed in older mice in respect of AD pathology [44], it is reasonable to assume that this may also be the case in respect of the HC of older Adrβ_3_KO mice. However, even though we could not find any differences regarding GFAP expression in the HC, we did find that the Adrβ_3_KO animals had increased EAAT2 expression in this nucleus. EAAT2 is the main transporter responsible for the reuptake of glutamate by astrocytes in the synaptic cleft [45]. Glutamate is the major excitatory neurotransmitter in the brain [45], but extracellular excess of glutamate increases the production of reactive oxygen/nitrogen species, which induces oxidative stress, leading to neuronal death [46]. EAAT2 regulates and buffers the amount of synaptic glutamate, preventing neuronal damage due glutamate excitotoxicity [47]. Even though the role of EAAT2 is well recognized, its response to different types of stress is complex and is still under investigation. In this respect, it has been shown that while stroke inhibits the expression of EAAT2, Adrβ-blocker attenuates this inhibition [48]), suggesting that adrenergic receptors have a role in the expression of EAAT2, corroborating our results. Interestingly, MS was unable to modify the expression of EAAT2 in the HC of older mice. 

Regarding the AMY, we did find an increase in GFAP expression in the AMY of the Adrβ_3_KO mice when compared to the WT mice. In addition, when the MS protocol was initiated later in life, there was no reduction in GFAP and the Iba-1 expression, as observed in the younger mice. Despite this, changes in the EAAT2 expression in the AMY of the Adrβ3KO mice when compared to WT mice were evident in the MS group. The majority of glutamate uptake is through EAAT2 [45]. Interestingly, lower EAAT2 expression or activity has been reported in several neurological disorders, such as amyotrophic lateral sclerosis [49], Alzheimer’s disease [50] and schizophrenia [51]. Given the evidence that reduced EAAT2 is associated with brain diseases, it could be hypothesized that MS protects against EAAT2 dysregulation, which may play a role in normal cognition [52,53], although this hypothesis is only speculative and additional studies are needed to better understand the mechanisms by which EAAT2 expression or activity could alter cognitive functions.

Although we observed an increase in GFAP expression in the AMY of older Adrβ_3_KO mice, a marker of the astrocyte activation, our data do not support Adrβ_3_ as an important adrenergic receptor mediating the anti-inflammatory effects of NE. Nevertheless, Adrβ_3_ may have a pivotal role in astrocytic modulation, possibly controlling the different roles of astrocytes, such as synaptic pruning and glutamatergic depuration, rather than inflammation. Further studies are needed to clarify the role of these receptors in respect of the functions of astrocytes. However, our data did show that this receptor is key to the cognition response in young and older mice.

A complex and dynamic MS protocol that exposed the animals to different kinds of stimuli, such as gustatory and olfactory stimuli of positive and negative valence, intellectual challenges to reach food, the use of hidden objects, and the presentation of food in ways that promoted foraging, rescued the memory deficit of young Adrβ_3_KO mice when applied immediately after weaning. It seemed to do this through decreasing neuroinflammation. 

It should be noted that the constant exposure of animals to novelty through the MS protocol used in the present study involved some level of stress to the animals. In addition, the animals were exposed to stimuli with negative valence, such as bedding with the smell of rats. Our results suggest, therefore, that a moderate level of stress experienced early in life could be beneficial for cognition. 

The observation that aging worsens the memory of Adrβ_3_KO mice when compared to the WT ones is notable despite the AMY activation caused by the valence of the stimulus. It has been shown that locus coeruleus (LC) degeneration is a common neuropathological feature of neurogenerative diseases, such as Alzheimer’s disease [54,55]. In fact, early degeneration of the LC could trigger, or be involved in, the progression of neurogenerative diseases [56,57,58,59]. The fact that Adrβ_3_KO inactivation leads to a greater loss in cognition with aging highlights the role of the noradrenergic signaling pathway in the course of dementia. 

In healthy rodents, LC projections to different brain regions begin to decline by 7–15 months of age [60,61]. Other studies with rodents and primates have found a correlation between memory loss and the progressive appearance of lesions, as well as consequent cell loss in the HC and entorhinal cortex, with age [62,63]. Advancing age leads to a loss of 10 to 20% of brain mass when compared to a young brain. This can lead to variations in cell loss in different brain regions and, consequently, more serious losses in certain regions than in others [64]. Adrβ_3_ inactivation, combined with the functional changes typical of advancing age, can aggravate damage to memory-formation processes, possibly explaining the worsening in memory observed in the adult Adrβ_3_KO mice. 

The MS protocol used in the present study did not affect locomotor capacity when applied to young animals, regardless of the genotype, but increased ambulatory activity in older Adrβ_3_KO mice. This suggests that the stimulus represented by MS may improve the activity of animals at an older age. The influence of MS on the exploratory behavior of mice has already been evaluated in other studies; however, as yet, there is no consensus on its influence [65,66]. 

An important limitation of the present study is that we only studied males. It is possible that Adrβ_3_KO female mice would respond differently to the MS, considering that the MS could exhibit a sex-dependent influence on cognition and behavior [67,68]. 

In conclusion, the results of the present study reinforce the idea that early stimulation of individuals is beneficial for cognition and can prevent or delay early memory impairment caused by defects in the neuronal signaling involved in cognition. They also showed that Adrβ_3_ has an important role in memory as aging worsened the memory of Adrβ_3_KO animals. The observed expression of glial cells and glutamate transporter supports the idea that changes in glial cells, especially the astrocytic response, may be an important component of adrenergic modulation and induced cognitive deficit in Adrβ_3_ knock-out (Adrβ_3_KO) mice, but further studies are required to prove this hypothesis.

## 4. Materials and Methods

Animals: Adrβ_3_KO mice (Mus musculus) with an FVB background were generated by removing the 306bp genomic fragment containing the sequences encoding the third to fifth transmembrane domains of Adrβ_3_, replacing it with a neomycin selection cassette, as described by Susulic et al. [20]. We purchased the animals from Jackson Laboratory (Bar Harbor, ME, USA) and established an in-house colony at the animal facility at Mackenzie Presbyterian University (Sao Paulo, Brazil). All mice used in this study were genotyped using RT-PCR to confirm their status as homozygous knock-out (Adrβ_3_KO) or wild-type (WT) mice. In total, 32 male Adrβ_3_KO mice and 29 male WT controls from different litters randomized between groups were used in a protocol approved by the Institutional Committee on Animal Research at the Center of Biological Sciences and Health, Mackenzie Presbyterian University (CEUA/UPM No. 156/02/2017). Mice were housed in groups at 26 °C, 55–60% humidity and a 12 h light/dark cycle with ad libitum access to standard food (Nuvilab, Paraná, Brazil) and water. In the current study, we focused on male mice to reduce the number of confounding factors, considering that the sex hormone fluctuation observed during the estrous cycle in females leads to changes in behavior and learning [69,70].

Experimental design: We evaluated the effects of multisensory Stimulation (MS) at two periods in the lives of the animals: (i) immediately after weaning on post-natal day (PND) 21 and lasting until PND 85; and (ii) in adult life, with MS starting on PND 120 and lasting until PND 180.

Effect of early MS on young mice: The animals were transferred immediately after weaning on PND 21 to a cage with two floors and were submitted to the protocol described in Table 1 until PND 85. Behavioral tests were then started and were completed on PND 120 (Figure 8A). The animals were divided into the following groups: WT (n = 7); Adrβ_3_KO mice (n = 7); WT MS (n = 9); and Adrβ_3_KO MS (n = 9).

Effect of late MS on adult mice: The animals were kept in standard housings until PND 120 when they were then transferred to the cage with two floors and submitted to the protocol described in Table 1 until PND 180 (Figure 1B). Behavioral tests were started on PND 180 and finished on PND 205. The animals were divided into the following groups: WT (n = 6); Adrβ_3_KO (n = 9); WT MS (n = 7); and Adrβ_3_KO MS (n = 7).

Multisensory stimulation protocol: All mice submitted to the MS remained in two-story housings (57 × 31 × 41 cm), lined with wood shavings and with a shelter, water and chow diet on both floors. The control groups remained in regular housing (a Plexiglas cage measuring 30 × 20 × 13 cm). The MS protocol used was that standardized in our laboratory with some adaptations [21,22] and consisted of two interventions per week for eight weeks in the morning, with sensory, cognitive and dietary activities to keep the novelty throughout the whole protocol (Table 1). After eight weeks of MS, the behavioral tests were started. During the behavioral assessment, the animals remained in the two-story housings until the completion of the tests, but without the stimulatory activities. 

Behavioral testing: All tests were performed in the morning (7:00–9:00 a.m.) under dimmed light (15 lux) and recorded by video for later analysis by two different blind observers in the following order for both studies 1 and 2. 

Open-field test (OF): The open-field test was used to evaluate locomotor and exploratory activity [71]. The animals were placed in the center of a circular acrylic arena (diameter = 30 cm), divided into four central zones and eight peripheral zones (Insight Ltd., São Paulo, Brazil), in a low-light environment (15 Lux) for 10 min. Locomotion (total number of lines crossed with all four paws) in the central and peripheral zones and time spent in the periphery were evaluated using the software OpenFLD v 1.0 (OpenFLD v1.0—available at http://blog.sbnec.org.br/2010/07/software-gratuitos-para-analise-do-labirinto-em-cruzelevado-e-campo-aberto/ accessed on 15 January 2018). Rearing was evaluated manually by two independent blind observers. The test was performed three consecutive times in a 24 h interval [72].

Novel object recognition test (NOR): This test was performed to evaluate short- and long-term memory. It was performed in the OF arena immediately after the OF test to guarantee the habituation of the mice to the arena. The test consists of three stages: familiarization, test (3 h later) and retest (24 h later). In the familiarization stage, the animals were placed in the open-field arena for 10 min and were then exposed to two unknown identical objects, object O1 and object O1′ for 3 min. Three hours later, the test was performed with the animals being placed in the arena for 3 min and exposed to object O1 and a new object (O2). Twenty-four hours after familiarization, the animals were placed in the arena for 3 min and exposed to the known object O1 and a new object (O3). At each stage, the time the animal spent exploring the object with their nose was expressed as a recognition index, i.e., the percentage of time spent with each object of the total time spent with both objects [73]. Time spent with each object was evaluated manually by two independent blind observers.

Social recognition test (SR): Social preference and discrimination were evaluated using a non-automated, three-chambered box with three successive and identical chambers (Stoelting, Dublin). The protocol used is similar to the one described previously by Moy et al. (2018) [74]. Briefly, in the familiarization period, the mice were allowed to explore the three chambers freely for 10 min starting from the intermediate compartment, with the two other chambers containing empty wire cups. To test social preference, the test mouse was placed in the intermediate compartment, while an unfamiliar mouse was now put in one of the wired cups in a random and balanced manner. The doors were re-opened, and the test mouse was allowed to explore the three chambers for 10 min. The time spent in each of the chambers, the number of entries into each chamber and the time spent sniffing each wired cup were recorded to measure social preference. In the third phase, social discrimination was evaluated with a new, unknown mouse being placed into the remaining empty wire cup with the test mouse allowed to explore the entire arena for 10 min, having the choice between the first, already-investigated mouse (known) and the novel unfamiliar mouse (unknown). The same measures were taken as for the social preference test [75,76]. Time spent with each cage was evaluated manually by two independent blind observers.

Western blot analysis: Immunoblotting was performed to analyze the expression of GFAP, Iba-1 and EAAT2 proteins with β-actin as an internal loading control according to the procedure described by Towbin et al. [77]. The HC and AMY were homogenized in 200 µL of radioimmunoprecipitation assay (RIPA) buffer [50 mM Tris, 150 mM NaCl, 1 mM EDTA, 0.1% SDS, 0.5% deoxycholate, and 1% NP-40] with a proteinase inhibitor cocktail (Thermo Fisher Scientific, Waltham, MA, USA) and centrifuged at 12,000× *g* for 15 min at 4 °C. The supernatant was collected and analyzed for protein concentration using the method of Bradford (Thermo Fischer Scientific, Waltham, MA, USA) [78]. The samples were diluted in Laemmli buffer and aliquots containing 30 μg of protein was loaded per lane and separated on 10% SDS-PAGE for GFAP and EAAT2 and on 15% SDS-PAGE for Iba-1, along with an unstained protein molecular-weight marker. The proteins were transferred electrophoretically to a nitrocellulose membrane (Bio-Rad Laboratories, Hercules, CA, USA) in a transfer buffer (25 mM Tris–HCl, 192 mM glycine and 20% (*v*/*v*) methanol, pH 8.3). The membranes were blocked for 1 h at room temperature with 5% (*w*/*v*) non-fat dry milk in Tris-buffered saline containing Tween (TBST: 25 mM Tris, pH 8.0, 150 mM NaCl and 0.05% tween 20), and then washed in TBST. The membranes were then probed with anti-GFAP, anti-Iba-1 and EAAT2 antibodies diluted at 1:500, 1:1000 and 1:2000, respectively, in 3% (*w*/*v*) TBST + 1% BSA + 0.1% Sodium Azide overnight at 4 °C. After washing with TBST, the membranes were incubated with the corresponding HRP-conjugated secondary antibodies diluted in 2% (*w*/*v*) non-fat dry milk in TBST. The membranes were washed three times with TBST, and the detection of proteins was carried out using a chemiluminescent kit (ECL, Amersham Biosciences, NJ, EUA). The target proteins were detected using a C-DiGit Western blot scanner (LI-COR, Lincoln, NE, USA). Densitometric analysis was conducted using ImageJ software (National Institutes of Health, USA).

### Statistical Analysis

Sample size: To determine the sample size, information from a pilot sample with twelve mice allocated into four groups (WT, AdrB3KO, WT MS, and AdrB3KO MS) was used. This showed that a minimum sample of five mice per group (20 mice in total) was necessary to detect differences in means at a significance level of 5% with 95% power in one-way ANOVA, regarding the differences in the percentage of time spent on each object (known and new). For this, the values of 153.13 and 127.20 were considered, respectively, for the mean square between groups and intragroup. 

The experimental data were analyzed using PRISM software (GraphPad Software, San Diego, CA, USA). The Shapiro–Wilk test was adopted to verify normality among data. For all analysis of the statistical significance of the differences between the mean values for the groups, two-way ANOVA was used, followed by Bonferroni’s multiple-comparisons test, and a *p* value < 0.05 was considered statistically significant.

## Figures and Tables

**Figure 1 ijms-24-10522-f001:**
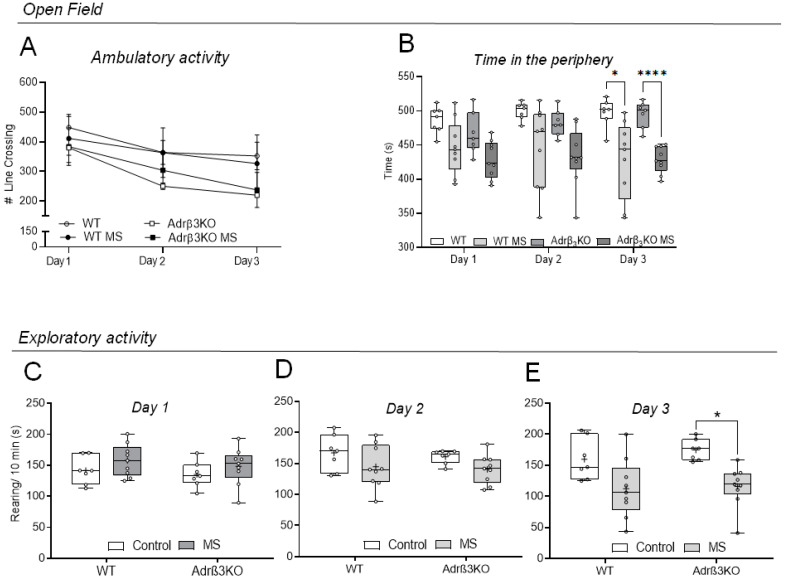
Effect of exposure to MS early in life on locomotor activity and anxiety behavior in young adult Adrβ_3_KO and WT mice. Open-field test. (**A**) All animals of all groups of young mice exhibited significantly less ambulatory activity during the days of observation (*p* = 0.025) with no difference among groups; (**B**) the MS protocol only produced a reduction in the time spent in the periphery in the OF in the WT group (* *p* = 0.038) and the Adrβ_3_KO group (**** *p* < 0.0001) on day 3 of observation. There was no difference in time spent in the periphery during days 2 and 3 of observations for all groups; (**C**–**E**) there was a reduction in the total number of rearings in the Adrβ_3_KO MS vs. the Adrβ_3_KO mice only on day 3 of observation (* *p* = 0.016) but not on day 1 or 2 of observation. There was no difference for WT on all 3 days of observation. All the results were analyzed using two-way ANOVA with Bonferroni’s post-hoc test. Values are expressed as median ± SE (**A**) or as median (25th percentile–75th percentile) (**B**–**E**) (WT n = 7; WT MS n = 9; Adrβ_3_KO n = 7; Adrβ_3_KO MS n = 9).

**Figure 2 ijms-24-10522-f002:**
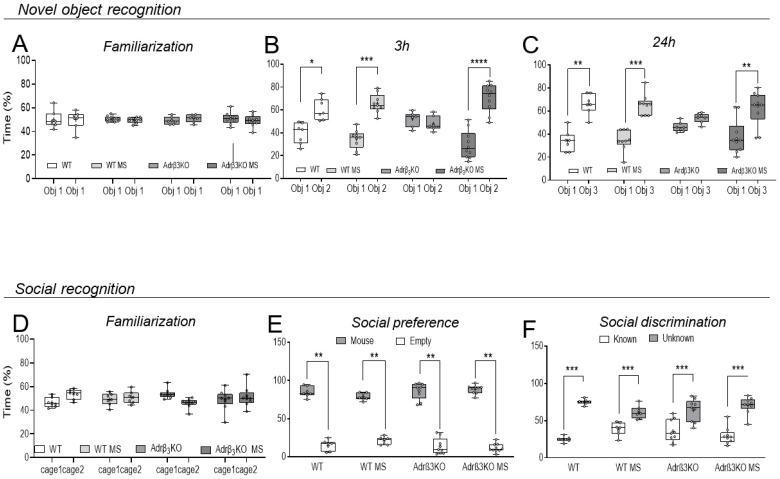
MS exposure early in life corrects cognitive impairment in young adult Adrβ_3_KO mice: (**A**) All groups explored the objects similarly during the familiarization period. (**B**) Three hours after object familiarization, WT mice (* *p* = 0.025), WT MS mice (*** *p* = 0.0001) and Adrβ_3_KO mice (**** *p* < 0.0001) exposed to MS early in life spent significantly more time with a novel object (O2) than a familiar object (O1), but Adrβ_3_KO mice not exposed to the MS protocol spent a similar amount of time with both O1 and O2 (*p* > 0.99). (**C**) 24 h after object familiarization both WT (*** *p* < 0.0006) and Adrβ_3_KO (** *p* < 0.0041) mice exposed to MS early in life spent significantly more time with a novel object (O3) than a familiar object (O1). SR test. (**D**) All groups explored the empty chambers equally during the familiarization period. (**E**) Both WT and Adrβ_3_KO mice showed normal preference for social interaction and spent significantly more time in the chamber with a mouse than with an empty cup, regardless of whether they had been exposed to MS (** *p* < 0.0001). (**F**) Both WT and Adrβ_3_KO mice, with or without MS, showed normal preference for social novelty and spent significantly more time in the chamber with an unknown mouse than in the chamber with the known mouse (*** *p* < 0.0001). The data were analyzed using two-way ANOVA, followed by Bonferroni’s multiple-comparison test. Values are expressed as median (25th percentile–75th percentile) (WT n = 7; WT MS n = 9; Adrβ_3_KO n = 7; Adrβ_3_KO MS n = 9.)

**Figure 3 ijms-24-10522-f003:**
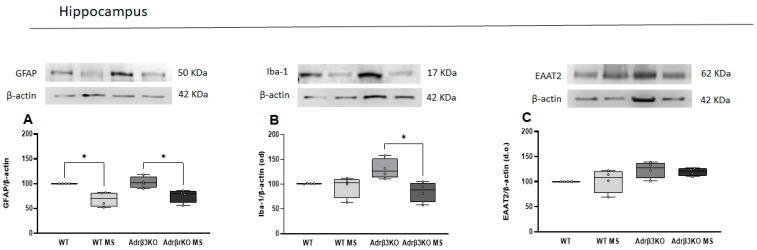
Expression of GFAP, Iba-1 and EAAT2 in hippocampus of young adult Adrβ_3_KO mice: (**A**) The MS protocol decreased the expression of GFAP in both WT (*p* = 0.011) and Adrβ_3_KO mice (* *p* = 0.027); (**B**) MS protocol decreased Iba-1 expression only in Adrβ_3_KO (* *p* = 0.004); (**C**) EAAT2 expression was not affected by genotype nor MS protocol. The data were analyzed by one-way ANOVA followed by Bonferroni’s post-hoc test. Values are expressed as median (25th percentile–75th percentile) (WT n = 4; WT MS n = 4; Adrβ_3_KO n = 4; Adrβ_3_KO MS n = 4.)

**Figure 4 ijms-24-10522-f004:**
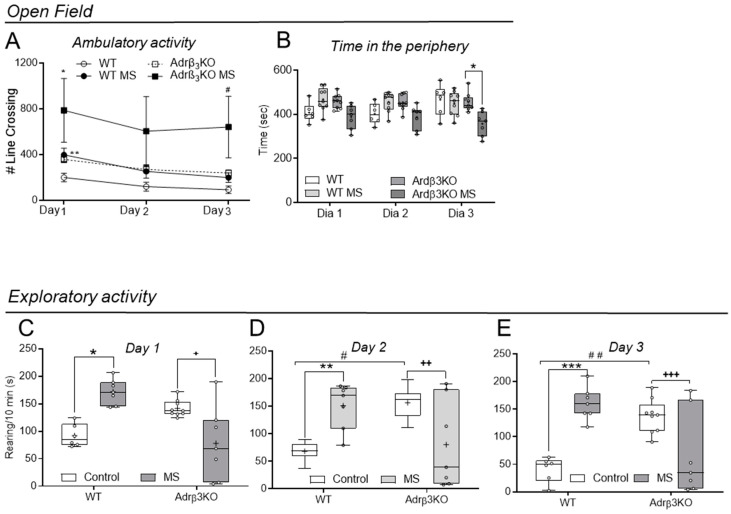
Effect of exposure to MS late in life on locomotor activity and anxiety behavior in young adult Adrβ_3_KO and WT mice. Open-field test. (**A**) Total number of line crossings of WT and Adrβ_3_KO mice without exposure to MS and with exposure to MS. On day 1 of observation, the ambulatory activity of Adrβ_3_KO mice was significantly higher when compared to WT mice (** *p* = 0.02). On day 1 of observation, the MS protocol increased ambulatory activity only in the Adrβ_3_KO MS group when compared to Adrβ_3_KO (* *p* = 0.037). On day 3 of observation, Adrβ_3_KO MS exhibited an increase in ambulatory activity when compared to WT MS (^#^ *p* < 0.02); (**B**) on day 3, the Adrβ_3_KO MS mice spent less time in the periphery than the Adrβ_3_KO group (* *p* = 0.015); (**C**–**E**) The total number of rearings increased in the WT MS when compared to the WT on day 1 (* *p* = 0.003), day 2 (** *p* = 0.02) and day 3 (*** *p* = 0.0003); The total number of rearings increased in Adrβ_3_KO mice when compared to the WT mice on day 2 (^#^
*p* = 0.003) and on day 3 of observation (^##^
*p* = 0.002). The total number of rearings decreased in the Adrβ_3_KO MS vs. Adrβ_3_KO adult mice on day 1 (^+^ *p* = 0.01), day 2 (^++^
*p* = 0.02) and day 3 (^+++^ *p* = 0.02). The data were analyzed by two-way ANOVA followed by Bonferroni’s post-hoc test. Values are expressed as median ± SE (**A**) or as median (25th percentile–75th percentile) (**B**–**F**) (WT n = 6; WT EE n = 7; Adrβ_3_KO n = 9; Adrβ_3_KO EE n = 7).

**Figure 5 ijms-24-10522-f005:**
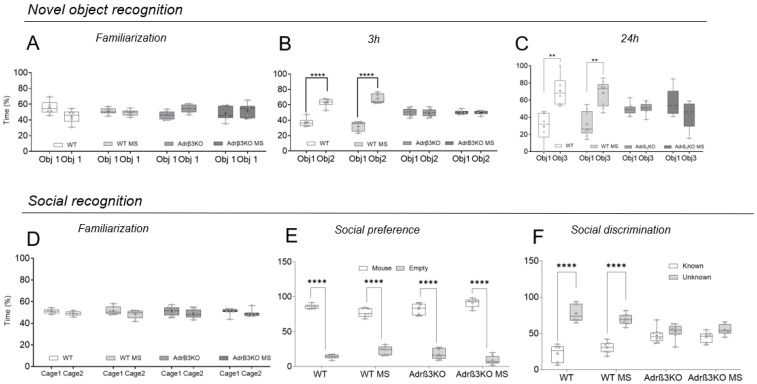
MS exposure late in life does not correct cognitive impairment in adult Adrβ_3_KO mice: novel object recognition test. (**A**) All groups explored the objects similarly during the familiarization period. (**B**) 3 h after object familiarization, the WT and WT MS mice spent significantly more time with a novel object (O2) than a familiar object (O1) (**** *p* < 0.0001), while the Adrβ_3_KO and Adrβ_3_KO MS mice spent an equal amount of time with both the O1 and O2 (*p* > 0.9999); (**C**) 24 h after object familiarization, the WT and WT MS mice spent significantly more time with a novel object (O3) than a familiar object (O1) (** *p* = 0.0040), while the Adrβ_3_KO and Adrβ_3_KO MS mice spent an equal amount of time with both O1 and O3 (*p* > 0.9999). Social recognition test. (**D**) All groups explored the empty chambers equally during the familiarization period. (**E**) All groups showed normal preference for social interaction and spent significantly more time in the chamber with a mouse than with an empty cup, regardless of exposure to MS (**** *p* < 0.0001). (**F**) Both WT (**** *p* = 0.004) and WT MS (** *p* = 0.002) mice showed normal preference for social novelty and spent significantly more time in the chamber with an unknown mouse than in the chamber with the known mouse, while Adrβ_3_KO mice, with or without MS, spent an equal amount of time with both known and unknown mice (*p* > 0.999). The data were analyzed using two-way ANOVA followed by Bonferroni’s post-hoc test. Values are expressed as median (25th percentile–75th percentile) (WT n = 6; WT MS n = 7; Adrβ_3_KO n = 9; Adrβ_3_KO MS n = 7).

**Figure 6 ijms-24-10522-f006:**
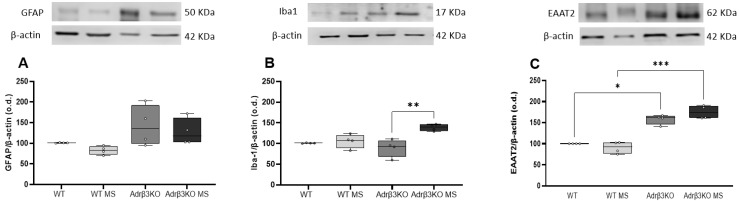
Expression of GFAP, Iba-1 and EAAT2 in hippocampus of older Adrβ3KO mice: (**A**) The expression of GFAP was not affected in older mice regardless of the genotype or the MS protocol; (**B**) Iba-1 expression was increased in Adrβ_3_KO MS when compared to Adrβ_3_KO mice (** *p* = 0.002); (**C**) EAAT2 expression was increased in Adrβ_3_KO mice when compared to WT mice (* *p* = 0.013) and in Adrβ_3_KO MS mice when compared to WT MS mice (*** *p* = 0.002). The MS protocol did not affect the expression of EAAT2, regardless of the genotype. The data were analyzed using one-way ANOVA followed by Bonferroni’s post-hoc test. Values are expressed as median (25th percentile–75th percentile) (WT n = 4; WT MS n = 4; Adrβ_3_KO n = 4; Adrβ_3_KO MS n = 4).

**Figure 7 ijms-24-10522-f007:**
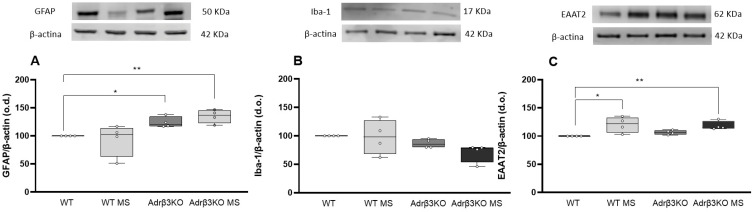
Expression of GFAP, Iba-1 and EAAT2 in amygdala of older Adrβ_3_KO mice: (**A**) The expression of GFAP was increased in Adrβ_3_KO mice when compared to WT mice (* *p* = 0.045), and in Adrβ_3_KO MS mice when compared to WT mice (** *p* = 0.031). The MS protocol did not affect the expression of GFAP regardless of the genotype; (**B**) Iba-1 expression was not affected by the MS protocol or by genotype; (**C**) EAAT2 expression was increased in the WT MS mice when compared to WT mice (* *p* = 0.02) and was increased in the Adrβ_3_KO MS mice when compared to WT mice (** *p* = 0.034) but was not affected by genotype. The data were analyzed using one-way ANOVA followed by Bonferroni’s post-hoc test. Values are expressed as median (25th percentile–75th percentile) (WT n = 4; WT MS n = 4; Adrβ_3_KO n = 4; Adrβ_3_KO MS n = 4).

**Figure 8 ijms-24-10522-f008:**
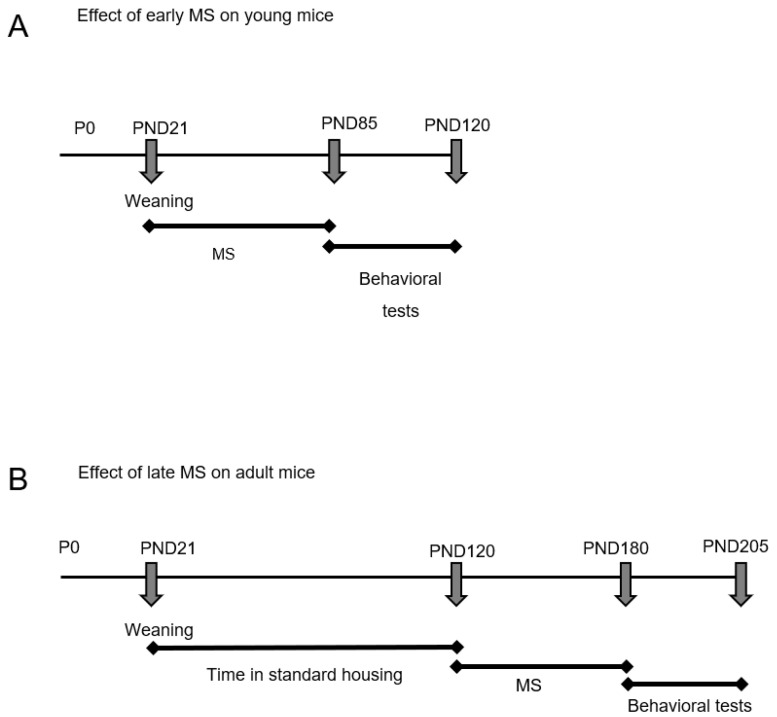
Experimental design. (**A**) Multisensory stimulation (MS) started immediately after weaning on post-natal day 21 (PND2 1) and continued until PND 85, when the behavior assessment was performed until PND 120. During the tests, the animals remained in stimulatory housing but without interventions. (**B**) MS was initiated in adult life on PND 120 and continued until PND 180, when the behavior assessment was performed until PND 205.

**Table 1 ijms-24-10522-t001:** Multisensory Stimulation Protocol.

Weeks	First Intervention	Second Intervention
1st	Familiarization with the new environment	Banana (100 g), apple (50 g), grape (25 g) for 5–6 h
2nd	Exposure to cotton balls of different sizes for 4–5 h	Hiding fruit under the bedding for 4–5 h
3rd	Exposure to ice with and without water for 1 h	Exposure to newspaper sheets for 5–6 h
4th	Exposure to carrots (100 g) in different sizes for 4–5 h	Exposure to plastic balls in a box filled with bedding for 3–4 h
5th	Exposure to trail with seasonings (oregano, lemongrass and chamomile) for 2 h	Exposure to cooked rice (50 g) for 3–4 h
6th	Exposure to two extra burrows made from cardboard	A banana (100 g) hanging from a thread attached to the roof of the housing for 2–3 h
7th	Exposure to mirrors for 15 min	Exposure to neutral jelly with raisins (8 units) inside for 3 h
8th	Exposure to bedding from a rat housing placed in four different locations in the housing for 1 h	Exposure to bowls containing water with frozen peas, carrots and corn for 2–3 h

## Data Availability

All statically significant results are shown in figures and tables.

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
