# Peer review of "Multisensory Stimulation Reverses Memory Impairment in Adrβ3KO Male Mice"

_ijms, 2023, doi:10.3390/ijms241310522_

Round 1
Reviewer 1 Report
The title should inform that all data was generated in male mice.
Line 82 – first mention of Open Field (OF).
Line 86 – EE or MS?
Line 114 – define EE – enriched environment? First mention in the text.
Line 128 – What is SR? Social recognition? First time appeared in the text, please define the abbreviation.
Line 139 – Adrb3KO animals
Figure 3 – Although the text refers to Figure 3 as animals that underwent MS protocol early in life, figure legend also indicates data on animals with MS intervention during adulthood. Authors should address that and make sure the Figure is clear on which data is presented.
In the result section “Ambulatory and exploratory activity of Adrβ3KO and WT mice exposed to MS late in life”, authors indicate dataset as Figure 4 but is actually Figure 5. Adjustments to subsequent Figures/datasets are also required.
Line 236: worse cognitive impairment
It seems Adrb3KO renders animals a higher baseline of exploratory activity
Figure 5D/E – Please add legend to # comparison (WT x Adrb3KO). Please keep consistency with group names (Adrb3)
The manuscript would greatly benefit from proofreading and/or editing to improve flow and readability.
Author Response
Reviewer #1
Thank you so much for your review of the manuscript and insightful comments.
- The title should inform that all data was generated in male mice.
- Thank you. We added the word male in the title.
Line 82 – first mention of Open Field (OF).
- Thank you. We added the full name.
Line 86 – EE or MS?
- Thank you. Initially, we called the protocol as Environmental Enrichment, but decided later that Multisensory Stimulation was more accurate to describe the intervention. We checked the manuscript, and all the “EE” were replaced by “MS”.
Line 114 – define EE – enriched environment? First mention in the text.
- Thank you. Please, see the answer above.
Line 128 – What is SR? Social recognition? First time appeared in the text, please define the abbreviation.
- Thank you. We added the Social Recognition test.
Line 139 – Adrb3KO animals
- Thank you. We corrected this mistake.
Figure 3 – Although the text refers to Figure 3 as animals that underwent MS protocol early in life, figure legend also indicates data on animals with MS intervention during adulthood. Authors should address that and make sure the Figure is clear on which data is presented.
- Thank you. We apologize for this mistake. We corrected the legend.
In the result section “Ambulatory and exploratory activity of Adrβ3KO and WT mice exposed to MS late in life”, authors indicate dataset as Figure 4 but is actually Figure 5. Adjustments to subsequent Figures/datasets are also required.
- Thank you. We corrected the number of the Figures.
Line 236: worse cognitive impairment.
- Thank you. We corrected this.
It seems Adrb3KO renders animals a higher baseline of exploratory activity.
- Thank you. The 2-way ANOVA analysis showed a p>0.999 for comparing WT and AdrB3KO in all three days of observation.
Figure 5D/E – Please add legend to # comparison (WT x Adrb3KO). Please keep consistency with group names (Adrb3).
- Thank you. We added the significance for # and corrected the group names.
Comments on the Quality of English Language
The manuscript would greatly benefit from proofreading and/or editing to improve flow and readability.
- Thank you. We sent the manuscript to a native English reviewer.

Reviewer 2 Report
The paper about the effect of multisensory stimulation in Adrβ3KO mice is actual and interesting but some points need to be clarified to make sure the conclusions are correct. Minor point:1. Not all abbreviations are deciphered at the first mention (MS, SR). Major points: 1. The age of animals during memory test was different between early and later MS groups. Did authors check that differences between Adrβ3KO with MS groups are related with age of MS start but not the age during memory test? 2. Authors wrote: “In total 32 male Adrβ3KO mice and 29 male WT controls from different litters randomized between groups were used “ and “In the current study, we focused on male mice to reduce the number of confounding factors”. It is necessary to note this limitation of investigation in Discussion section because effect of additional stimulation may have sex-dependent differences. For example, it was shown by J.A. Cosgrove and co-authors (2022) the sex-dependent influence of postweaning environmental enrichment in Angelman syndrome model mice (https://doi.org/10.1002/brb3.2468). Also it is necessary to enumerate clearly the confounding factors.
Author Response
Reviewer #2
Thank you so much for your review of the manuscript and insightful comments.
The paper about the effect of multisensory stimulation in Adrβ3KO mice is actual and interesting but some points need to be clarified to make sure the conclusions are correct.
Minor point:
- Not all abbreviations are deciphered at the first mention (MS, SR).
- Thank you. We corrected this mistake.
Major points:
- The age of animals during memory test was different between early and later MS groups. Did authors check that differences between Adrβ3KO with MS groups are related with age of MS start but not the age during memory test?
- Thank you. We did not evaluate the memory of older animals (180 days of age) with MS started early in life. It is possible that older mice under MS protocol since 21 days of age would exhibit a memory impairment with aging despite the stimulation, but we did not address this question. We added this limitation of the study in the Discussion.
- Authors wrote: “In total 32 male Adrβ3KO mice and 29 male WT controls from different litters randomized between groups were used “ and “In the current study, we focused on male mice to reduce the number of confounding factors”.
It is necessary to note this limitation of investigation in Discussion section because effect of additional stimulation may have sex-dependent differences. For example, it was shown by J.A. Cosgrove and co-authors (2022) the sex-dependent influence of postweaning environmental enrichment in Angelman syndrome model mice (https://doi.org/10.1002/brb3.2468).
- Thank you. The confounding factors are essentially the fluctuation of the sex hormones during the estrous cycle of the female mice. It is described in the literature that the exploratory and locomotor activity of the female rodent changes along the estrous cycle phases what can be considered as an important confounding factor. When the observation is a long term one, the estrous cycle is not as important, but if the behavior test is performed in 1-3 days, it can impact the results.
Also it is necessary to enumerate clearly the confounding factors.
- Thank you. We added this in the Discussion session.
Round 2
Reviewer 2 Report
The authors answered my questions, I have only minor comments:
1. What does the "+" sign inside the 25th percentile–75th percentile mean in figure 5 C, D (may be mean value?) and some other pictures?
2. It is necessary to increase and align the font in Figure 6 and others so that they are the same.
Author Response
The authors answered my questions, I have only minor comments:
Thank you for continuing to review the manuscript.
1. What does the "+" sign inside the 25th percentile–75th percentile mean in figure 5 C, D (may be mean value?) and some other pictures?
A. Thank you. Yes, it is the mean value.
2. It is necessary to increase and align the font in Figure 6 and others so they are the same.
A. Thank you. We aligned the increased font in Figure 6.